# Effects of Acute Caffeine Ingestion on Cognitive Performance before and after Repeated Small-Sided Games in Professional Soccer Players: A Placebo-Controlled, Randomized Crossover Trial

**DOI:** 10.3390/nu15143094

**Published:** 2023-07-10

**Authors:** Rodrigo Freire de Almeida, Mateus de Oliveira, Isadora Clivatti Furigo, Rodrigo Aquino, Neil David Clarke, Jason Tallis, Lucas Guimaraes-Ferreira

**Affiliations:** 1Centre of Physical Education and Sports, Federal University of Espirito Santo, Vitória 29075-910, ES, Brazil; rodrigofalmeida80@gmail.com (R.F.d.A.); mateus.p.oliveira10@hotmail.com (M.d.O.); rodrigo.aquino@ufes.br (R.A.); 2Centre for Health and Life Sciences, Coventry University, Coventry CV1 5FB, UK; isadora.clivattifurigo@coventry.ac.uk; 3Centre for Physical Activity, Sport and Exercise Sciences, Coventry University, Coventry CV1 5FB, UKjason.tallis@coventry.ac.uk (J.T.)

**Keywords:** soccer, caffeine, Stroop test, executive function, nutritional supplementation, small-sided games

## Abstract

Soccer is a team sport that requires players to process a significant amount of information quickly and respond with both speed and accuracy to the ever-changing demands of the game. As such, success in soccer depends not only on physical attributes but also on cognitive abilities such as perception and decision-making. The aim of the current study was to investigate the acute effects of caffeine ingestion on Stroop test performance before and after repeated small-sided games (SSG) in professional soccer players. Twelve professional male soccer players (29 ± 4.1 years; 78.1 ± 7.7 kg body mass) participated in this study. A randomized crossover double-blind placebo-controlled trial was used. Caffeine (5 mg.kg^−1^) or a placebo was ingested 45 min before a protocol consisting of five 5 min SSG with 1 min rest intervals. A computerized version of the colour Stroop test was completed immediately before and after the exercise protocol. During the Stroop test, words appeared on the computer screen in three different ways: (i) neutral words (neutral condition); (ii) correspondent colour (i.e., “red” painted in red; congruent condition), or; (iii) different colour (i.e., “red” painted in green; incongruent condition). The incongruent condition aimed to cause the interference effect, as the colour and the word did not match. Ratings of perceived exertion (RPE) were assessed after each SSG. RPE increased during the five sets of the SSG protocol (*p* < 0.001), without differences between the caffeine and placebo trials. The soccer-specific exercise protocol promoted a faster response during the Stroop test (two-way ANOVA main effect for SSG protocol: *p* < 0.05), with no differences in accuracy (*p* > 0.05). Caffeine ingestion resulted in slower reaction time during the Stroop test during the congruent and neutral trials but not during the incongruent trial (two-way ANOVA main effect for supplementation: *p* = 0.009, *p* = 0.045, and *p* = 0.071, respectively). Accuracy was lower in the caffeine trial in congruent and incongruent trials (*p* < 0.05 caffeine vs. placebo both on the pre- and post-SSG protocol). In conclusion, a soccer-specific exercise protocol improved the Stroop test performance in professional soccer players, but acute caffeine ingestion (5 mg.kg^−1^) was detrimental.

## 1. Introduction

Executive function encompasses a range of cognitive abilities that are essential for goal-directed behaviour. These cognitive processes include working memory, planning, problem-solving, sustained attention, inhibition, and cognitive flexibility. [1]. Among several neuropsychological tests, the Stroop test is often used for experimental and clinical purposes [2], assessing selective attention and the ability to inhibit cognitive interference [3]. Previous studies showed that an acute bout of low- or moderate-intensity exercise improved cognitive performance during the Stroop test. For example, Yanagisawa et al. [4] showed that young adults improved their Stroop test performance and presented high dorsolateral prefrontal activation after an acute aerobic exercise session at moderate intensity (50% of VO2max). Similarly, Alves et al. [5] demonstrated that an acute bout of aerobic or resistance exercise can improve performance during the Stroop test in healthy women, demonstrating that different modalities of exercise may result in cognitive performance improvement. Soccer is an open-skill team sport where the players need to deal with a large amount of information in a short time, responding with both speed and accuracy to the quick-changing game demands, which is associated with the often-called “soccer IQ”, or “game intelligence” [6]. Previous investigations have demonstrated that executive functions are associated with success in soccer players [7,8], indicating that cognitive performance enhancers have the potential to improve performance in soccer. 

Dietary supplements are used by athletes at all levels of sport with the ultimate aim of improving performance and recovery [9]. Caffeine (1,3,7-trimethylxanthine) is a stimulant present in foods, beverages, and medicines and consumed by approximately 85% of adults in the U.S. [10,11]. It is one of the most consumed nutritional supplements in sports, aiming for improvements in cognitive and physical performance [12,13]. After ingestion, caffeine reaches its peak in plasma after 45 to 60 min, with a half-life of around 3–5 h [14]. Caffeine has been used to improve physical performance in soccer [15,16,17,18] but investigations on the effects of caffeine ingestion on executive function in the general population have been presenting contrasting results [19,20,21,22]. The available research on the effects of caffeine supplementation on cognitive function in athletes is limited and yields conflicting findings [23,24,25,26]. These discrepancies might be attributed to variations in the specific athletic populations studied as well as differences in the mode and dosage of caffeine administration.

For example, Hogervorst et al. [24] examined the effects of caffeine supplementation on cognitive function in 24 well-trained male cyclists. The participants underwent a 2.5-h exercise protocol at 60% VO2max, followed by a time-to-exhaustion trial. They consumed either a drink containing 100 mg of caffeine and 45 g of carbohydrate, a carbohydrate-only beverage, or a placebo. The results revealed that, compared to the carbohydrate-only placebo conditions, the ingestion of the caffeinated beverage led to improved response times during the Stroop colour-word test when compared to noncaffeinated beverages. In contrast, Russell et al. [26] conducted a study involving 14 professional rugby union players to explore the impact of caffeine-gum chewing (400 mg of caffeine) on physical and cognitive performance, utilizing the Stroop test. The results indicated that cognitive performance did not exhibit any significant differences when compared to the performance of participants who chewed noncaffeinated gum. In another study, Ali et al. [25] examined the effects of caffeine administration (6 mg.kg^−1^) in female team-game players. The authors reported a tendency towards faster response times during the Stroop test, although the results did not reach statistical significance (*p* = 0.072). Consequently, additional studies are warranted to gain a deeper understanding of the impact of caffeine on executive function in athletes, particularly by using sport-specific exercise protocols.

Another important aspect to consider pertains to the doses required to elicit positive effects on both physical and cognitive performance. Currently, there is ambiguity regarding whether the doses of caffeine necessary to enhance performance are equivalent to those needed for improved cognitive function. Furthermore, it remains uncertain whether higher doses of caffeine have the potential to enhance physical performance while simultaneously impairing cognitive function. To the best of our knowledge, the effects of acute caffeine ingestion on executive function in soccer players were not investigated. Therefore, the aim of the present study was to examine the effects of consuming a moderate dose of caffeine (5 mg.kg^−1^) on the performance on the Stroop test in professional soccer players, both before and after engaging in a soccer-specific exercise protocol using small-sided games (SSG). Our hypothesis was that pre-exercise caffeine ingestion improves cognitive performance before and after a soccer-specific exercise protocol.

## 2. Materials and Methods

### 2.1. Participants

Twelve professional soccer players participated in this study (29 ± 4.1 years; 78.1 ± 7.7 kg body mass). All players were recruited from a professional soccer club in Brazil. All participants were active in soccer training, with no injuries or illnesses, and were familiar with the procedures herein. Exclusion criteria included injured or returned from injury, utilization of psychoactive or psychodepressive drugs, and presence of Daltonism. Goalkeepers were excluded from the analysis since their behavioural pattern differs from other players. The procedures followed the principles outlined in the Declaration of Helsinki and were approved by the University Ethics Committee for Study in Humans (registration 66849317.9.0000.5542). The participants signed an informed consent form after a verbal explanation about the procedures used in the study.

### 2.2. Procedures

A crossover randomized double-blind placebo-controlled trial was used in the current study. Two weeks after the familiarization session, participants performed an SSG exercise protocol after the ingestion of caffeine (5 mg.kg^−1^) or placebo solutions. Seven days later, the SSG protocol was repeated, totalling two experimental sessions using a placebo or caffeine in a randomized manner. Cognitive performance was assessed by the Stroop test 45 min after caffeine or placebo ingestion and immediately after the exercise protocol. The experimental design is illustrated in Figure 1.

Testing sessions were performed at the same time of the day (08 am) to avoid variations in the circadian cycle. Environmental conditions were: (a) temperature (°C): start: 31.9 ± 1.98, end: 32.45 ± 3.21; (b) relative air humidity (%): start: 51.85 ± 5.31, end: 51.3 ± 5.36; (c) wind speed (m/s): start: 4.35 ± 4.46, end: 3.08 ± 5.91. These variables were measured using a digital thermo-higro-anemometer (Akrom KR825, São Leopoldo-RS, Brazil).

The SSG was composed of a GK + 3 × 3 + GK format (3 players + goalkeeper on each team) with a constant area of 36 m × 27 m and 5 min duration. After 5 min of warm-up, five series of SSG were performed, with 1 min of recovery between each SSG. Team formations and experimental conditions (caffeine and placebo) were randomized. Each team consisted of a goalkeeper (not used in the analysis), a defender, a midfielder and a forward. The team formation was the same for both experimental trials and all players of each team were under the same condition (caffeine or placebo). No specific verbal instructions were provided before, during, or after the SSGs, which followed official soccer rules, including offside. Water was consumed ad libidum for hydration. The objective of each game was to score and outscore the opponents. Ratings of perceived exertion (RPE) were assessed after each SSG using a 0–10 scale (from “no exertion at all” to “maximal effort”) [27]. A previous study with soccer players demonstrated that RPE correlates with heart rate and blood lactate during SSG, being a valid measurement of global exercise intensity in soccer [28]. 

Caffeine ingestion: 45 min before testing, participants ingested 5 mg.kg^−1^ of caffeine (Sigma-Aldrich, Sydney, FL, USA) diluted in 250 mL of artificially sweetened water or a placebo drink consisting of 250 mL of artificially sweetened water alone. After ingesting the capsules, participants remained comfortably seated until the start of the testing protocol. Caffeine doses ranging from 3 to 6 mg.kg^−1^ are often used to improve physical performance [14], so the dose in the current study was chosen to investigate if it is also capable of improving cognitive performance when associated with a soccer-specific exercise protocol. The solutions were identical in flavour and colour. The participants were asked to have the same breakfast on test days and to abstain from nutritional supplements and caffeine-containing foods and drinks 48 h before testing sessions. The blinding protocol was assessed using a model adapted from Klauss et al. [29] consisting of two questions: do you think you received/are receiving treatment? Regardless of the answer, yes or no, a second question was asked: how confident is your impression? This last question contains a Likert scale from 1 to 5 (1: none; 2: little; 3: average; 4: much; and 5: extreme). Five participants (41%) correctly identified the use of caffeine and the Likert scale on how confident participants were about the treatment received resulted in a median value of 3 for both conditions. Habitual consumption of caffeine was obtained through the questionnaire based on information from Maughan [30]. The average daily caffeine intake was 139 ± 85 mg per day. 

Stroop test: the computerized colour Stroop test consists of the visualization of coloured words on the computer screen (i.e., red, blue, yellow, and green) and the identification of the correct colour using the corresponding keyboard key, which were painted in the respective colour. Participants were asked to select the correct colour of the word, not the word itself, as quickly as possible. Words appeared on the computer screen in three different ways: (i) neutral words (neutral condition); (ii) correspondent colour (i.e., “red” painted in red; congruent condition), or; (iii) different colour (i.e., “red” painted in green; incongruent condition). The incongruent condition aimed to cause the interference effect, as the colour and the word did not match. The Stroop test is a valid tool to assess selective attention and the ability to inhibit cognitive interference [2,3]. The test was applied using the Psychology Experiment Building Language (PEBL) version 2.1 with 48 stimuli of 1 s of duration for each Stroop test condition [31]. For all tests, the participants used the same hardware to avoid the influences of the computer’s processing speed. The test duration was ~5 min. Data were extracted from software and the mean response time (ms) and accuracy (% of correct responses) were determined for neutral, congruent, and incongruent conditions.

### 2.3. Statistical Analysis

Data normality was confirmed using the Shapiro–Wilk test. Therefore, the results were presented as mean ± standard deviation (SD). RPE and Stroop test performance (response time and accuracy) were assessed using a 2-way repeated measures analysis of variance using the Geisser–Greenhouse epsilon correction for violations of the sphericity assumption. The independent variables used in the analysis were: condition (placebo vs. caffeine); and time (from SSG1 to SSG5 in the RPE assessment and pre- vs. post-SSG protocol on the Stroop test analysis). Post hoc analysis using Tukey’s multiple comparisons test was performed where any significant interaction and main effects were found. A *p*-value of 0.05 was used to establish statistical significance. The statistical analysis was performed using the software GraphPad Prism, version 8.0.

## 3. Results

The RPE was assessed after each SSG. RPE increased during the five SSG protocols (ANOVA main effect for time: F2.242;24.66: 16.55; *p* < 0.001), with no differences between caffeine and placebo conditions (ANOVA main effect for supplementation: F1;11: 1.590; *p* = 0.23; interaction main effect: F2.307;25.38: 1.54; *p* = 0.23, Figure 2). Post hoc analysis indicated that RPE increased after the fourth and fifth SSGs after caffeine ingestion and after the fourth SSG after placebo ingestion (*p* < 0.05; Figure 2). 

When the response time during the Stroop test was analysed, significant ANOVA main effects were observed at congruent (F1;11 6.64; *p* = 0.009; Figure 3A) and neutral conditions (F1;11: 6,79; *p* = 0.045; Figure 3C), with slower response during the caffeine condition. The SSG protocol resulted in a faster response during all Strop test trials (SSG ANOVA main effect *p* = 0.006; *p* = 003; and *p* = 0.005 during the congruent, incongruent, and neutral trials, respectively). On the other hand, the soccer-specific protocol did not result in changes in Stroop test accuracy in all trials (SSG ANOVA main effect *p* = 0.0.057; *p* = 0.782; and *p* = 0.662, during the congruent, incongruent, and neutral trials, respectively).

In the congruent condition, caffeine ingestion resulted in lower accuracy compared to the placebo (caffeine main effect: F1;11: 6.97; *p* = 0.023; Figure 3D) with no effect of the exercise protocol (F1;11: 4.538; *p* = 0.057). In the incongruent condition, a significant interaction was found (interaction main effect: F1;11: 5.99; *p* = 0.032, Figure 3D), although Tukey’s post hoc test did not show significant pre vs. post differences in both caffeine and placebo conditions (*p* > 0.05). No significant ANOVA main effects were observed for caffeine (F1;11: 3.66; *p* = 0.082) or the exercise protocol (F1;11: 0.080; *p* = 0.78; Figure 3E). No differences were found in accuracy during the neutral condition of the Stroop test (*p* > 0.05 for all ANOVA main effects, Figure 3F).

## 4. Discussion

The current study aimed to investigate the effects of acute caffeine ingestion on Stroop test performance before and after a soccer-specific exercise protocol in professional soccer players. The results indicated an increased RPE during five bouts of SSG for both the placebo and caffeine conditions. The exercise protocol resulted in a faster Stroop test response in all conditions with no significant effect on accuracy. Caffeine ingestion resulted in a slower response during the Stroop test in the congruent and neutral conditions, as indicated by the condition (placebo vs. caffeine) ANOVA main effects and lower accuracy in the congruent and incongruent conditions (*p* < 0.05 for placebo vs. caffeine both pre- and postexercise protocol). Therefore, our initial hypothesis that caffeine ingestion would improve cognitive performance before and after performing a soccer-specific exercise protocol was rejected. To the best of our knowledge, this is the first study that has investigated the effects of caffeine ingestion and soccer-specific exercise on cognitive performance in soccer players.

As mentioned, RPE increased throughout the repeated SSG protocol but no differences were observed between the placebo and caffeine conditions. This contrasts with findings from a meta-analysis suggesting that acute caffeine consumption reduces RPE during exercise [32]. Nonetheless, the results indicate that the disparities in cognitive performance between the placebo and caffeine conditions are not attributable to caffeine’s impact on RPE. To assess the players’ cognitive function in the present study, a computerized version of the Stroop test was used. John Ridley Stroop originally developed the test in 1935 to evaluate the capacity for selective visual attention and cognitive inhibition [3]. Its purpose is to measure the ability to suppress automatic responses. The Stroop effect, or interference effect, is supported by evidence showing an asymmetry; words interfere with colour naming but the reverse does not happen. This indicates that reading words is a more automatic process in terms of brain processing compared to naming colours [33]. Soccer is an open-skill team sport that demands reactions to external information to take the appropriate decisions and actions in a very quick manner. Executive functions, such as attention and the ability to inhibit cognitive interference, assessed with the Stroop test, are needed for high performance in soccer. For example, Vestberg et al. [7] evaluated the executive function using the D-KEFS test battery from the highest Swedish soccer leagues and from lower leagues and found that the executive functions can predict success in the sport, since the players from higher leagues outperformed those from lower leagues, and the results in the executive test were significantly associated with performance in game statistics (goals and assists) during the competitive season. Thus, it is possible that changes in the executive function observed in the present study in response to caffeine ingestion have implications for the player during the game, even though this is considerably speculative.

It was also demonstrated that mental fatigue, as induced by the continuous Stroop test application, can result in physical and technical performance impairment. Smith et al. [34] demonstrated that mental fatigue induced by 30 min of the Stroop test resulted in impaired physical and technical performance in soccer players assessed through the yo-yo intermittent recovery test 1 (Yo-Yo IR1) and the Loughborough passing and shooting tests, respectively. Based on this, interventions aiming to improve cognitive performance or attenuate cognitive decline during exhaustive exercise can be important for sports that involve open skills and decision-making during the game, such as soccer. In this regard, caffeine has been studied as a valid ergogenic aid due to its actions on the central nervous system (CNS) and positive effects on physical performance [12,13,18]. 

Caffeine crosses the blood–brain barrier by diffusion due to its hydrophilic and lipophilic characteristics [35]; thus, it can easily reach all areas in the CNS. The mechanism by which caffeine ingestion may influence the state of alertness, arousal, and RPE is the inhibition of adenosine receptors. Adenosine is a purine with the general function of inhibiting neural activity, reducing locomotor activity, increasing the perception of effort, and inhibiting the release of neurotransmitters and hormones [36]. Since caffeine acts as a competitive inhibitor of adenosine, increasing its concentration in the CNS results in facilitating the release of noraepinephrine, dopamine, acetylcholine, serotonin, and GABA [37]. The result is the reduction of pain perception, increased alertness and spontaneous locomotor activity, increase in the rate of firing of motor units in active muscles, and attenuation of fatigue [35,38]. Possibly the first work investigating the effects of caffeine ingestion on Stroop test performance was published in 1989 by Foreman et al. [19]. It was demonstrated that the ingestion of 250 mg of caffeine resulted in impaired Stroop test performance, with slower responses when compared to noncaffeinated drinks. That observation aligns with the results from the current work where, after caffeine ingestion, participants exhibited a slower response (in the congruent and neutral conditions) and less accuracy (in the congruent and incongruent conditions). 

In female team-sports players, caffeine (6 mg.kg^−1^) promoted positive effects on vigour, but the effects on Stroop test performance after intermittent exercise were not statistically significant [25]. Other work corroborated those results, with no effects of caffeine ingestion on Stroop test performance in fresh participants [21,39,40]. On the other hand, Patat et al. [22] demonstrated that a single dose (600 mg) of a slow-release form of caffeine resulted in a faster response during the Stroop test in sleep-deprived subjects. This observation may suggest that caffeine can exert positive cognitive performance in fatigated individuals. Additionally, Hogervorst et al. [24] demonstrated that the ingestion of 100 mg of caffeine before a cycling protocol resulted in faster responses during the Stroop test in trained cyclists, with no effect on accuracy during the test. The substantially higher caffeine dosage used in the present study (5 mg.kg^−1^, which corresponds to a 400 mg dose for an 80 kg individual) is commonly used to improve physical performance. This suggests that higher doses of caffeine can possibly elicit detrimental effects during the Stroop test, as also demonstrated by Foreman et al. [19]. The observed findings can be attributed to the impact of varying levels of arousal on cognitive function, with the effects of caffeine on arousal appearing to depend on the dosage administered. Lower doses of caffeine can enhance positive mood and potentially alleviate anxiety, while higher doses can lead to increased tension and symptoms of anxiety, including nervousness and jitteriness [14]. The Yerkes–Dodson law states that performance improves with higher levels of arousal up to a point and then decreases in an inverted U shape [41]. While the effects of caffeine on arousal levels are documented [42], the impact of dosage on cognitive performance is less understood. It is possible that lower caffeine doses can promote a favourable arousal state, resulting in positive cognitive performance, while higher doses may be harmful. Additionally, the effects of caffeine ingestion might be influenced by the arousal level of the participant prior to caffeine ingestion [14,43]. Hence, due to the highly individualized response to caffeine, it becomes challenging to provide a general recommendation for caffeine dosage. Consequently, an individualized supplementation regimen may be required to determine the optimal caffeine dose for achieving the best cognitive performance.

In the current study, acute caffeine ingestion (5 mg.kg^−1^) resulted in a slower and less accurate response during the Stroop test in professional soccer players. As the relation between caffeine dosage and cognitive performance may exist, possibly with significant interindividual differences to its action, the results herein should not be extrapolated to lower caffeine dosages. However, as higher doses (i.e., 5–6 mg.kg^−1^) are commonly used to improve physical performance by players, the current study investigated its effects on cognitive performance. In the present study, the athlete’s hydration status was not monitored and this represents a noteworthy limitation. It is well-established that hydration status can impact cognitive function, including visual perceptual abilities [44]. However, it is worth emphasizing that the athletes had unrestricted access to water during the intervals of the repeated SSG protocol, and their *ad libitum* water consumption was similar to what they typically have during regular field training sessions. Another limitation of the present study is the assessment of cognitive performance with the Stroop test, which is significantly different from the cognitive demands observed during a soccer game. Scharfen and Memmert [45] emphasized that successful performance in real game situations requires the integration of both sport-specific perception (i.e., cognitive functions) and action (i.e., motor skills). Therefore, strategies aimed at improving soccer performance must address both aspects simultaneously. De Almeida et al. [46] aimed to address this gap in knowledge by utilizing a soccer-specific tactical analysis protocol, the system of tactical assessment in soccer (FUT-SAT), to investigate the effects of caffeine intake on tactical performance in professional soccer players. The results of the study showed that caffeine ingestion had mixed effects on tactical performance, with improvements seen in some aspects, while others worsened or showed no change. More studies using methodologies assessing tactical parameters and/or decision-making during soccer games or soccer-related activities will shed light on the potential value of caffeine ingestion as an ergogenic aid to improve both physical and cognitive performance for soccer players.

## 5. Conclusions

In the present study, it was observed that engaging in a soccer-specific exercise protocol involving repeated SSG resulted in improved response times across all trials of the Stroop test, regardless of the presence of cognitive interference. However, acute ingestion of 5 mg.kg^−1^ of caffeine had a negative impact on the performance of professional soccer players in the computerized Stroop test. Although this dosage has shown effectiveness in some exercise protocols, caution should be exercised when considering moderate to high doses of caffeine due to the significance of cognitive aspects during soccer matches. Further research is needed to explore the relationship between initial arousal levels and caffeine dosage on cognitive responses in athletes, as well as potential variations in individual responses to the same caffeine dose, such as those influenced by genetic factors or habitual caffeine consumption.

## Figures and Tables

**Figure 1 nutrients-15-03094-f001:**
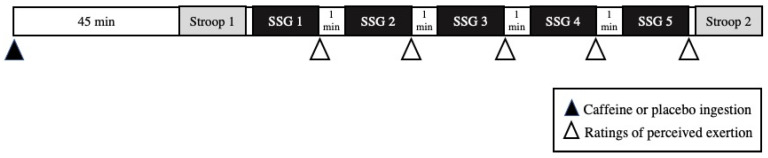
Experimental protocol. SSG: Small-Sided Game (5 min each). Stroop: Computerised Colour Stroop test.

**Figure 2 nutrients-15-03094-f002:**
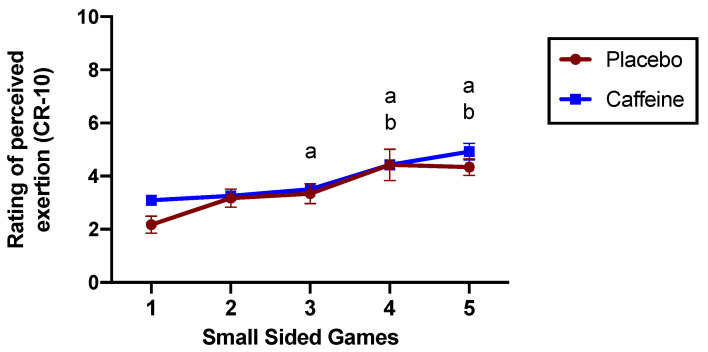
Rating of perceived exertion during 5 small-sided games in placebo and caffeine condition. Condition (placebo vs. caffeine) main effect: *p* = 0.23; SSG protocol main effect: *p* < 0.001; Interaction main effect: *p* = 0.23. Tukey’s multiple comparisons test: a = *p* < 0.05 vs. SSG 1 within placebo condition; b = *p* < 0.05 vs. SSG 1 within caffeine condition.

**Figure 3 nutrients-15-03094-f003:**
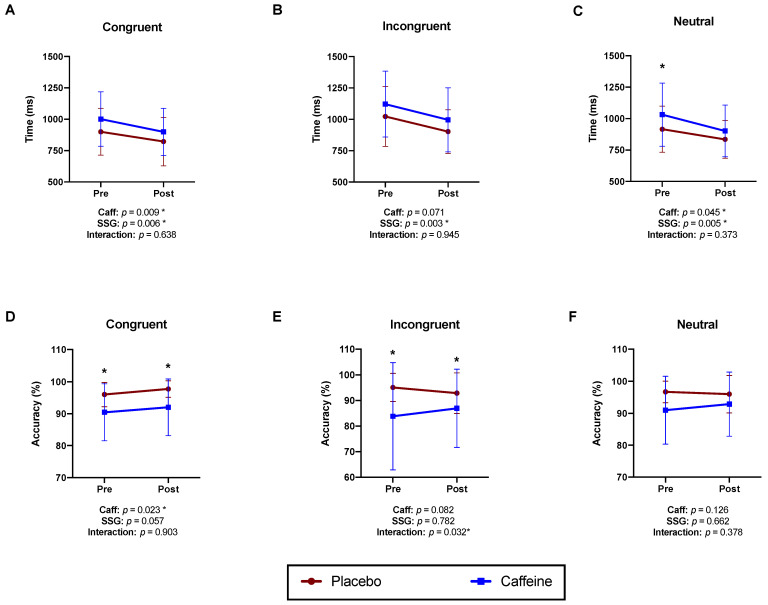
Stroop test performance before and after 5 small-sided games in Placebo and Caffeine conditions. Stroop test response time in congruent (**A**), incongruent (**B**) and neutral (**C**) conditions. Stroop test accuracy in congruent (**D**), incongruent (**E**) and neutral (**F**) conditions. Tukey’s multiple comparisons test: * = *p* < 0.05 caffeine vs. placebo within the same time point.

## Data Availability

The data presented in this study are available on request from the corresponding author.

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
