# Peer review of "Effects of Acute Caffeine Ingestion on Cognitive Performance before and after Repeated Small-Sided Games in Professional Soccer Players: A Placebo-Controlled, Randomized Crossover Trial"

_nutrients, 2023, doi:10.3390/nu15143094_

Round 1

Reviewer 1 Report

In this paper the authors report the results of a randomized control trial of caffeine ingestion on cognitive performance of professional football players. The authors find that, if anything, caffeine ingestion is detrimental.

The results of this study are likely to be of interest to readership of the journal. However, I have some concerns about the current analyses and interpretation of results. I also think that the presentation/clarity of the paper could be improved.

Major Comments

Abstract: The results of the congruent, neutral and incongruent components of the Stroop test are presented without context. For readers not familiar with the format of the test, these results will be hard to understand.

Section 2.2: The following aspects of the design are not clear. 1) How many trials was each participant exposed to (i.e., one in each intervention, or more)? 2) Was the order of placebo vs caffeine randomised? 3) Were there more than 1 participants in the same SSGs, or was there only ever one participant in each SSG? If there was more than 1, were they all on caffeine/placebo at the same time?

Section 2.2: RE the Stroop test, is the correct identification the color or the word written? For those not familiar with the test, this is not clear from what is described.

Section 2.3: These data are non-independent, because of the cross-over design. It is not clear how the paired nature of the data was handled (if at all). I also concerned about the use of ANOVA. The RPE data are, in reality, ordinal, though I can just about see how one could justify ANOVA. However, the accuracy data will be binomial and need to be analysed as such. For example, the SDs in Fig 3 span 100% accuracy, indicating the data are close to the upper boundary.

Figure 2: It looks like the wrong figure legend has been included here.

Section 3: The written results RE the Stroop test are inconsistent with the figure, and what is described in the discussion and abstract. For example, “Only in the neutral condition, Stroop Test response time was increased after the SSG protocol …”. And “At incongruent conditions, no differences were observed in response time to caffeine ingestion or the SSG protocol”. However, Figure 3 suggests improvements following SSG in all conditions. This is also what is concluded in the discussion.

L223-225: The nature of the interaction needs to be described.

Minor Comments

L61-64. This Is a long sentence, which would benefit from being broken up and re-written.

L74: “Caffeine HAS …”.

L75: “… on executive function in THE general population have …”.

I have made some suggestions for improvements.

Author Response

Thank you for your time and for agreeing to review our study.

Major Comments

Abstract: The results of the congruent, neutral and incongruent components of the Stroop test are presented without context. For readers not familiar with the format of the test, these results will be hard to understand.

Thank you for the comment. A brief description of each stroop test component was added to the abstract (highlighted in yellow) so reader will be able to better understand the differences between each trial.

Section 2.2: The following aspects of the design are not clear. 1) How many trials was each participant exposed to (i.e., one in each intervention, or more)? 2) Was the order of placebo vs caffeine randomised? 3) Were there more than 1 participants in the same SSGs, or was there only ever one participant in each SSG? If there was more than 1, were they all on caffeine/placebo at the same time?

Thank you for the comment, we agree that more details were needed to clarify the experimental design. We have added more information about the experimental design and procedures, all highlighted in yellow in the manuscript.

Section 2.2: RE the Stroop test, is the correct identification the color or the word written? For those not familiar with the test, this is not clear from what is described.

We have rephrased the explanation on the paragraph where the Stroop test is described. We believe it is clearer now for the reader.

Section 2.3: These data are non-independent, because of the cross-over design. It is not clear how the paired nature of the data was handled (if at all). I also concerned about the use of ANOVA. The RPE data are, in reality, ordinal, though I can just about see how one could justify ANOVA. However, the accuracy data will be binomial and need to be analysed as such. For example, the SDs in Fig 3 span 100% accuracy, indicating the data are close to the upper boundary.

Thank you for your feedback. We acknowledge the caveat, but currently, we are unaware of a non-parametric alternative to the two-way ANOVA for repeated measures, which is why we employed it in our study. To investigate whether there would be any discrepancies in the outcomes and interpretation of the article, we conducted paired t-tests between the caffeine and placebo conditions, both pre and post, and overall, the results remained consistent with those obtained from the 2way ANOVA analysis.

For example, using T-tests, it is still possible to find a shorter response time after caffeine ingestion in the congruent and neutral trials, but not in the incongruent one. Most analyses also show faster response after the exercise protocol in all Stroop Test trials.

Similiarly, caffeine resulted in lower accuracy in congruent and incongruent conditions, but not in neutral. Therefore, the results and general interpretation of the article would not be different when using this approach. If the reviewer has any other suggestions, we are open to hearing them and thank you again for your constructive comments.

Figure 2: It looks like the wrong figure legend has been included here.

Thank you for the observation. Legend was incorrect indeed. This have been changed now in the manuscript.

Section 3: The written results RE the Stroop test are inconsistent with the figure, and what is described in the discussion and abstract. For example, “Only in the neutral condition, Stroop Test response time was increased after the SSG protocol …”. And “At incongruent conditions, no differences were observed in response time to caffeine ingestion or the SSG protocol”. However, Figure 3 suggests improvements following SSG in all conditions. This is also what is concluded in the discussion.

Thank you for the accurate observation. We have changed the results section for correct presentation based on the results. We have also changed the conclusion for better clarity.

L223-225: The nature of the interaction needs to be described.

We have added more information in the manuscript, highlighted in yellow.

 Minor Comments

L61-64. This Is a long sentence, which would benefit from being broken up and re-written.

L74: “Caffeine HAS …”.

L75: “… on executive function in THE general population have …”.

Thank you for the comments above. Modifications were made and are highlighted in yellow in the manuscript.

Reviewer 2 Report

The paper is  interesting expecially  because it is dedicated to  the  cognitive  function impairment  in elite athletes  with potential injuries events in case  of detrimental cognitive status .  For this  reason  the  authors should report some  information  of  the  hydration in the pre - post phase  of  the  training session? The dehydration, especially  if calculated  by BIA analysis, could be  an important  component  of  the  decrease  of the  attention in athletes . In case  the  data  are avaliable , you can  discuss the  missing data and eventually  cite  some litterature's reports . 

 The  discrepancies found  with RPE  data  coulb be better  interpreted. 

Author Response

Thank you for your time and for agreeing to review our study.

The paper is  interesting expecially  because it is dedicated to  the  cognitive  function impairment  in elite athletes  with potential injuries events in case  of detrimental cognitive status .  For this  reason  the  authors should report some  information  of  the  hydration in the pre - post phase  of  the  training session? The dehydration, especially  if calculated  by BIA analysis, could be  an important  component  of  the  decrease  of the  attention in athletes . In case  the  data  are avaliable , you can  discuss the  missing data and eventually  cite  some litterature's reports . 

Indeed, your observation is very pertinent, and we agree that the hydration status might influence the cognitive function of athletes and could have been measured. Since we didn't do that control, we can't add that information. However, agreeing with your observation, we added a few lines about this aspect, pointing out the non-monitoring of the athletes' hydration status as an important limitation of the study.

The  discrepancies found  with RPE  data  coulb be better  interpreted. 

Thank you for your comment. We have added a brief discussion regarding RPE in our study (lines 267-272, highlighted in yellow in the manuscript).

Round 2

Reviewer 2 Report

The paper has been modifica sufficieny